# Effect of Experimental Aging on the Performance of Al-7.9Mg-13.3Zn and Al-10.5Mg-8.6Zn Alloys as Sacrificial Anodes

**Aline Hernández** [1,2,*]**, Álvaro Torres** [3]**, Sergio Serna** [4]**, Jan Mayén** [5] **and Bernardo Campillo** [1,6]

[1] Facultad de Química-UNAM, Circuito de la Investigación Científica S/N, Mexico City 04510, Mexico
[2] Facultad de Ingeniería, Universidad Anáhuac Mexico Norte. Avenida Lomas Anáhuac 46, Col. Lomas Anáhuac, Huixquilucan 52786, Mexico
[3] Facultad de Ciencias Químicas e Ingeniería P.A. Ing. Mecánica, Universidad Autónoma del Estado de Morelos, Av. Universidad 1001, Col. Chamilpa, Cuernavaca 62209, Morelos, Mexico
[4] CIICAp, Universidad Autónoma del Estado de Morelos, Av. Universidad 1001, Col. Chamilpa, Cuernavaca 62609, Mexico
[5] CONACYT, CIATEQ, Unidad San Luis Potosí, Eje 126 No. 225, Zona Industrial, San Luis Potosí 78395, Mexico
[6] Instituto de Ciencias Físicas-UNAM, Av. Universidad 1001, Col. Chamilpa, Cuernavaca 62609, Mexico
* Correspondence: alinehernandezgarcia@gmail.com or aline.hernandez@anahuac.mx; Tel.: +01-55-5627-0210 (ext. 7458)

**Abstract:** Light aluminum alloys have a great importance in industry owing to generally accessible costs, low density, good machinability, and corrosion resistance under certain environments. The present work studies aging treatments that preform important roles on the distribution and microstructural changes of two AlMg-Zn alloys, and the resulting effect on the corrosion behavior. The experimental AlMg-Zn alloys were cast and then heat treated at 200 °C, after the solubilization treatments were made, using different treatment times. These alloys showed important changes in their corrosion mechanisms, but mainly, corrosion started at $Al_xMg_yZn_z$ complex phases in both alloys. The optimal corrosion rates were reached after 5 and 24 h of heat treatment. These results were obtained through electrochemical techniques in NaCl solutions, and by metallographic analysis using SEM and optical microscopy.

**Keywords:** artificial aging; new aluminum alloys; sacrificial anodes

## 1. Introduction

Aluminum is one of the most important metals worldwide, and its alloys are highly used in today's engineering and industrial applications. According to Das. [1], 45 million tonnes of aluminum are produced annually, where 31% of the production corresponds to recycled aluminum. As a result, aluminum is the most recycled material and the second most used metal in the world [1]. Specific and expensive aluminum alloys (e.g., those alloyed with Hg, Ga, Sn, and In) are used in the cathodic protection industry [2]. The Al-Zn type, alloyed mainly with Hg (mercury) and In (indium), is the most efficient anode for cathodic protection against corrosion of structures exposed to marine environments. It is commonly accepted that aluminum corrosion resistance is mainly due to a crystalline oxide layer formation [3]. However, most electrochemical research is focused on the corrosion behavior of unalloyed aluminum in NaCl solutions. The success of such aluminum anodes is that both Hg and In prevent the formation of a continuous adherent and protective oxide film on the surface alloy, thus enabling continuous galvanic activity of the aluminum [4].

AlMg-Zn based alloys have raised strong attention since the 1980's. They are widely used in aerospace applications and manufacturing of high-speed boats and submarines due to the unique combination of lightweight and high mechanical properties [5]. The main corrosion form of this alloy system in seawater and NaCl solutions is pitting [6,7]. The AlMg-Zn alloy is characterized by a very heterogeneous microstructure, consisting of an aluminum solid solution matrix and various intermetallic phases; their mechanical properties are due to the presence of these particles [8]. The outstanding importance of the microstructure and the influence of the intermetallic particles on the corrosion behavior were extensively discussed by Campestrini et al. [9] and Vander Kloet [10]. Different local corrosion processes such as pitting corrosion, crevice corrosion or intergranular corrosion can be enhanced by the presence of intermetallic particles with cathodic characteristics, given the existence of a galvanic coupling with the aluminum matrix. This process produces a local increase in the pH, giving rise to the dissolution of the oxide layer in the area surrounding the intermetallic particle. Once this layer has been dissolved, the local alkalinity causes an intense attack on the interface between the matrix and the particles, as well as a detachment of the particles from the pit. Thereby, the presence of intermetallic particles with cathodic characteristics is the origin of pitting corrosion of aluminum-magnesium alloys. Previous research has described this phenomenon, where the reduction of oxygen occurs as the cathodic reaction on the intermetallic particles [11]. Parallel to the cathodic reaction, the anodic reaction is necessary in order for the passive layer to grow on the matrix and the thickening of this layer. On the other hand, Al matrix reaction with chloride ions can also evolve corrosion products surrounding the intermetallic particles [12]. However, Barbucci et al. [13] proposed the AlMg-Zn alloy as a promising alloy system to be studied for cathodic protection of structures exposed to marine environments, due to its low electrode potential, high current capacity, and the absence of Hg and In, which might pollute the sea [13]. More recently, it was reported that the presence and amount of Mg in Al alloys is important for cathodic protection effect, since it is the most active metal in the galvanic series and will always be the active anode when it is in contact with other metals [14]. The AlMg-Zn alloy system has a relatively complex equilibrium diagram. The first investigation of the entire system was carried out by Eger in 1913 [15]. From then on, AlMg-Zn alloys have been widely studied due to their excellent mechanical properties reached after age hardening [16]. Age hardening AlMg-Zn alloys show a combination of low density and high strength, and as a result have become the primary material used in aircraft and automotive industries. Gonzalez et al. [17] published that the magnesium in AlMg-Zn alloys played an important role on the $\tau$ ($Al_2Mg_3Zn_3$) phase particle distribution in $\alpha$-Al solid solution. This distribution can promote a good surface activation of the anode, avoiding the formation of the continuous, adherent, and protective oxide film on the alloy surface once it is in use.

This research aims to describe the electrochemical corrosion behavior in two as-cast AlMg-Zn alloys with increased dispersion of the $\tau$ phase in the matrix, through several aging treatments, that withdraw the fast, kinetic reactions occurring in solid state at 200 °C, after the solubilization treatments. Additionally, the effects of Mg addition on the $\tau$ phase distribution in the microstructure and galvanic efficiency of the AlMg-Zn alloys were investigated by SEM and optical microscopy.

## 2. Materials and Methods

Two AlMg-Zn alloys were designed by lowering and increasing the Mg content from a basic AlMg-Zn alloy. The alloys were melted by induction from commercially pure (99.5%) materials and poured in green sand molds. The contents of the alloying elements were analyzed using the ICP method. Results of the chemical analyses are shown in Table 1. Zn content was also varied in order to observe its effect on the resulting microstructures.

**Table 1.** Chemical compositions of the Al-Zn-Mg alloys in wt.%.

| Alloy | Mg | Zn | Si | Fe | Al |
|---|---|---|---|---|---|
| Base | 9.43 | 7.41 | 0.115 | 0.34 | Bal. |
| Series 1 | 7.93 | 13.32 | 0.119 | 0.40 | Bal. |
| Series 2 | 10.49 | 8.61 | 0.119 | 0.38 | Bal. |

Cylindrical samples of 1 cm$^2$ × 1 cm height were obtained from the casting. The homogenizing heat treatment was carried out at 430 °C during 17 h, quenched in liquid nitrogen. The age-heat treatments were performed in each alloy at 200 °C for 5, 24, and 100 h, respectively.

The as-cast and aged samples were sectioned longitudinally at mid-width using a diamond disc cutter (Weiyi, Qingdao, China). One side was prepared for microstructural characterization by grinding it on SiC metallographic paper, than polishing with 0.5 and 0.05 μm alumina power, and etched with Keller's and Graff-Sargent reagents (Keller's reagent: 95 mL water, 2.5 mL $HNO_3$, 1.5 mL HCl,1.0 mL HF, Graff and Sargent's reagent: 84 mL water, 15.5 mL $HNO_3$, 0.5 mL HF, 3 g $CrO_3$). Since the $Mg_2Si$ phase dissolved when using the Keller reactant, thus making it difficult to quantify, two chemical attack reactants were used instead. Microstructures were studied using an optical inverted research metallographic microscope model PMG3 (Olympus, Tokio, Japan) and a JEOL scanning electron microscope (SEM, Tokyo, Japan) using backscattered and secondary electron imaging operated at 20 kV. The SEM was equipped with an in situ energy dispersive spectrometer (Oxford Instruments Group, Osney, United Kingdom).

The electrochemical behavior of all aluminum-alloys was tested in a 3.5% NaCl solution. The electrochemical assays were carried out in a three-electrode cell arrangement. The aluminum alloy samples were put in a sample holder, presenting an exposed area of 1 cm$^2$ to the electrolyte. A platinum gauge was used as a counter electrode and a saturated calomel electrode was employed as a reference electrode. Double loop and polarization curves were carried out from 100 mV in the cathodic side to 1000 mV in the anodic side. The scan rate used was 1.003 mV/s.

After the electrochemical test, all the samples were studied a second time for microstructural characterization using SEM to observe and identify the different corrosion products formed on their surfaces. The identification of every aluminum alloy sample is illustrated in Table 2.

**Table 2.** Identification of the samples.

| Heat Treatment Condition (h) | 0 | 5 | 24 | 100 |
|---|---|---|---|---|
| Series 1 | S10 | S15 | S124 | S1100 |
| Series 2 | S20 | S25 | S224 | S2100 |

## 3. Results and Discussion

### 3.1. Microstructural Characterization Using Optical Microscopy

The averages of each phase present, its morphology, and distribution were analyzed for the microstructural characterization of the samples.

It is worth mentioning that two different types of etchants were used. The $Mg_2Si$ phase was more sensitive due to its size and distribution in the alloy from series 1. Hence, the attack to the samples of this phase was more aggressive and afterwards, the phase was completely diluted. Figure 1 shows the different reagents employed on the samples and the identification of the phases detected therein.

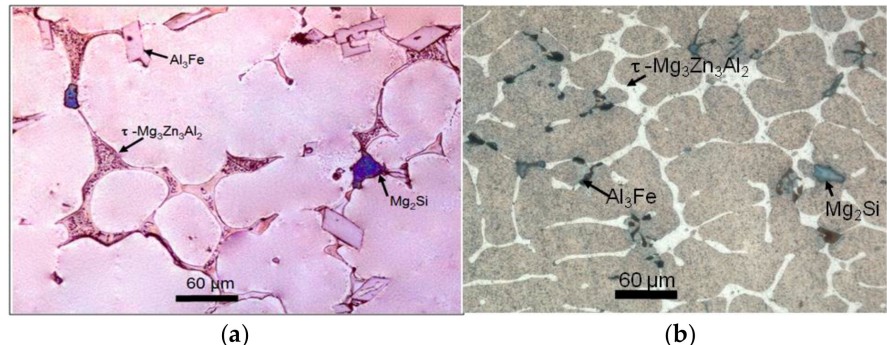

(**a**)                                                    (**b**)

**Figure 1.** Images of the samples with the different attacks employed: (**a**) Keller, (**b**) Graff-Sargent. Detected phases are identified.

In Figure 1, predominant phases found in this type of alloys are listed, such as τ-AlMgZn (Al$_2$Mg$_3$Zn$_3$) phase, Mg$_2$Si phase, Al$_3$Fe phase, and α-solution (matrix). Phase identification was performed through EDS analysis, using an EDS detector coupled to the scanning electron microscope. The obtained microanalysis is shown in Figures 2 and 3.

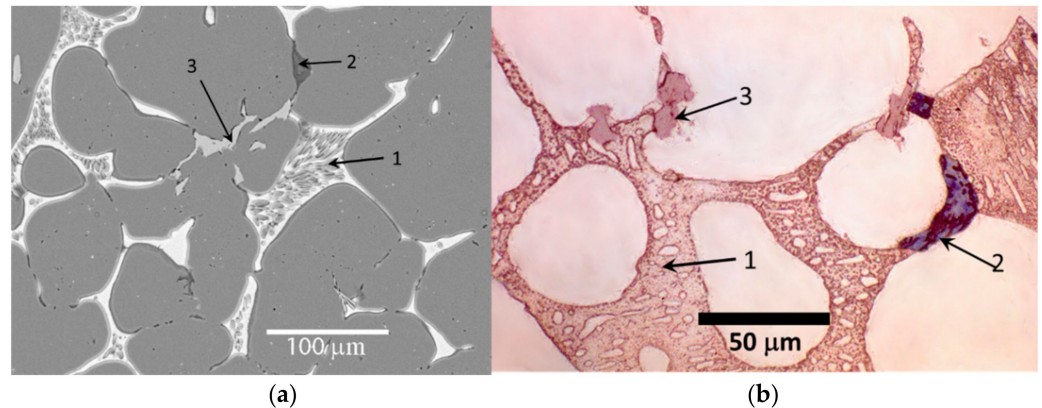

(**a**)                                                    (**b**)

**Figure 2.** Identification of phases: (1) τ-AlMgZn (Al$_2$Mg$_3$Zn$_3$), (2) Mg$_2$Si, (3) Al$_3$Fe; by (**a**) images obtained from scanning microscopy (200×) and (**b**) image from optical microscopy (500×).

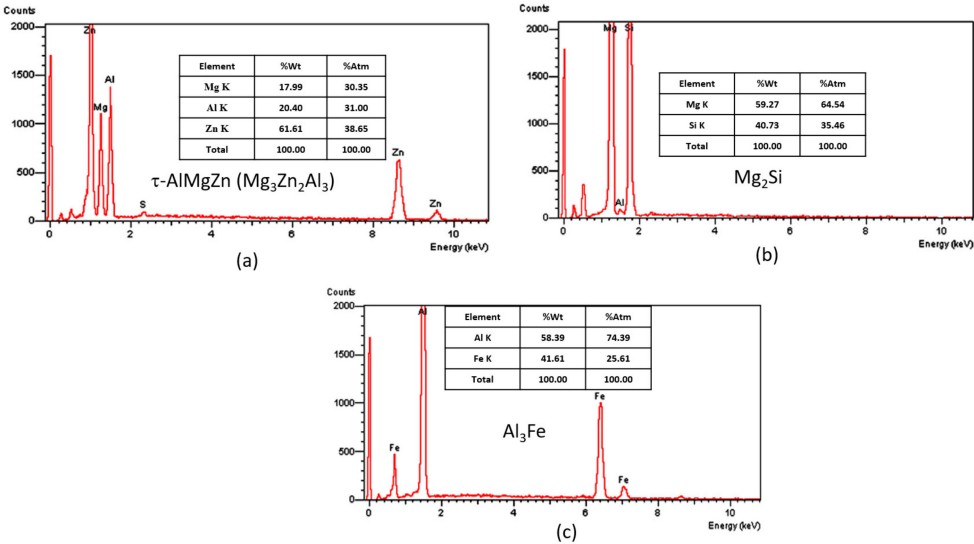

**Figure 3.** Microanalysis obtained. (**a**) τ-AlMgZn (Al$_2$Mg$_3$Zn$_3$), (**b**) Mg$_2$Si, and (**c**) Al$_3$Fe.

According to the microanalysis, the phases that contain the elements with the greatest reduction potential are $Mg_2Si$ and $\tau$-AlMgZn ($Al_2Mg_3Zn_3$), followed by the $\alpha$-matrix, and lastly the $Al_3Fe$ phase (Figure 3). The latter has a more positive reduction potential, due to the presence of Fe. Hence, with this microstructural distribution, the existence of galvanic couples may be expected.

The $Mg_2Si$ phase is closely attached to the $\tau$-AlMgZn phase. Similarly, the $\tau$-AlMgZn phase nucleates at the grain boundaries of the $\alpha$-matrix. The $\tau$-AlMgZn phase is homogeneously distributed throughout the matrix, while the $Mg_2Si$ phase is formed in the borders of the $\tau$-AlMgZn phase, i.e., where the coupling of the borders between the grains of the $\alpha$-solution are closer. The $Al_3Fe$ phase is also formed on the grain boundaries of the $\alpha$-matrix but is more dispersed in comparison to the other phases.

The general behavior observed is that the phases tend to decrease as the time of application of the treatment increases (Figures 4 and 5).

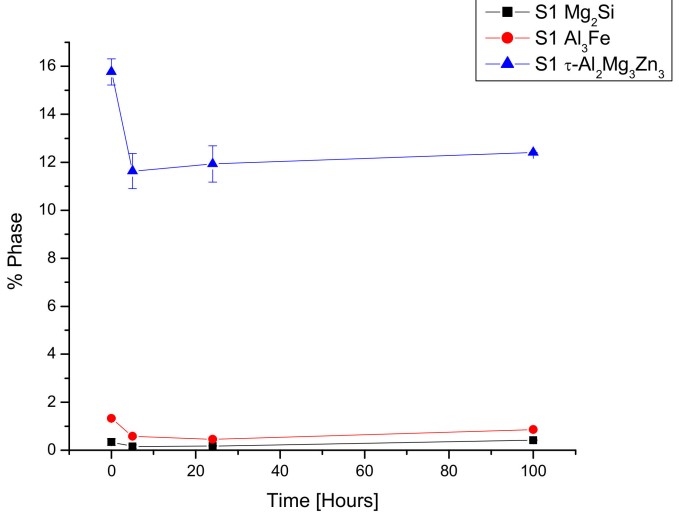

**Figure 4.** Phase % as a function of treatment time for samples in series 1.

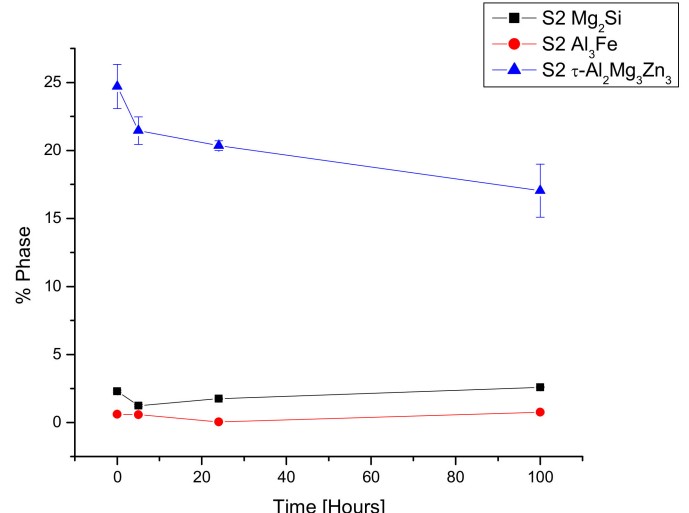

**Figure 5.** Phase % as a function of treatment time for samples in series 2.

As time increases, the number of phases present in the samples decreases. In both series, the phase in lower proportion is $Mg_2Si$; therefore, this phase is formed at the end of the solidification process. By applying the treatment process to the samples in series 1 (see Figure 4) for 5 h, the amount of the $Mg_2Si$ phase increases while the other two main phases decrease. This behavior arises when enough thermal energy is supplied, and the Mg tends to spread. As the $Mg_2Si$ phase is closely attached

to the $\tau$-AlMgZn phase (Figure 2), this phase loses the Mg, which in time enriches the Mg$_2$Si phase. This occurs in series 1, since it has a more homogeneous phase distribution and smaller phase sizes than the samples of series 2 (Figure 5), while the Zn and Al diffuse into the matrix. By increasing the time of the heat treatment, this pattern is refrained, and the main alloying elements tend to spread in the matrix. Accordingly, the decreasing of the phases occurs as a general trend, as the time of heat treatment increases. Tables 3 and 4 show the statistical quantifications of the phases in each sample.

**Table 3.** Phase quantification of samples of series 1 (% phase).

| Phase | As-Cast | 5 h | 24 h | 100 h |
|---|---|---|---|---|
| Al$_3$Fe | 1.32 | 0.58 | 0.45 | 0.86 |
| Mg$_2$Si | 0.34 | 0.15 | 0.17 | 0.41 |
| $\tau$-AlMgZn | 15.76 | 11.63 | 11.93 | 12.40 |

**Table 4.** Phase quantification of samples of series 2 (% phase).

| Phase | As-Cast | 5 h | 24 h | 100 h |
|---|---|---|---|---|
| Al$_3$Fe | 0.61 | 0.57 | 0.03 | 0.75 |
| Mg$_2$Si | 2.29 | 1.23 | 1.74 | 2.58 |
| $\tau$-AlMgZn | 24.69 | 21.45 | 20.34 | 17.04 |

The range of the percentages of the Al$_3$Fe phase is between 0.03% and 1.32% of the whole sample; the Mg$_2$Si phase range varies between 0.15% and 2.58%, while the $\tau$-AlMgZn phase is between 11.63% and 24.69%. This means that the predominant phase is $\tau$-AlMgZn, while the phase in the lowest proportion is Al$_3$Fe. The percentages of the phases formed in the different samples are consistent with the percentages of the alloying elements present (Table 1). Since the samples from series 2 have more Mg than the ones from series 1, the tendency to form more Mg compounds, i.e., $\tau$-AlMgZn and Mg$_2$Si, is natural. As the Mg is the alloying element with the highest reduction potential ($-2.37$ V), it is expected that the phases with Mg react more to the heat treatments, and likewise to the electrochemical tests. In series 1, the $\tau$-AlMgZn phase decreased 21.34%, the Mg$_2$Si phase 55.66%, and the Al$_3$Fe phase 56.03%. In series 2, the $\tau$-AlMgZn phase decreased 13.40%, the Mg$_2$Si phase 46.43%, and the Al$_3$Fe phase 88.16%.

*3.2. Microhardness*

Microhardness was evaluated to the samples, to determine their behavior depending on the different microstructures produced. The values are shown in Table 5.

**Table 5.** Vickers microhardness.

| Sample | Microhardness MHV (Series 1) | Microhardness MHV (Series 2) |
|---|---|---|
| Base | 115 | 115 |
| As-Cast | 180 | 168 |
| 5 h | 152 | 164 |
| 24 h | 145 | 144 |
| 100 h | 114 | 123 |

The micro-hardness values shown in Table 5 tend to decrease with the heat treatments. By modifying the amounts of Mg and Zn in the base alloy, the hardness is significantly increased; but as the aging treatments are applied, they decrease. The 100-h treatment time sample of series 1 reached a value similar to the base alloy. These trends can be seen in Figure 6.

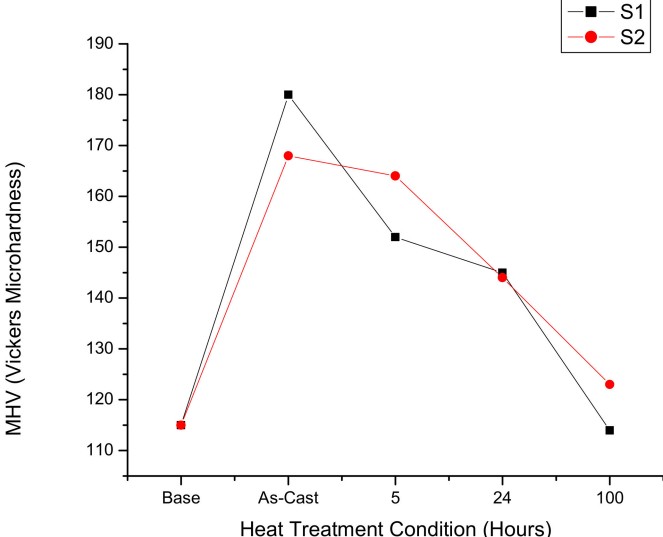

**Figure 6.** Vickers micro-hardness vs. heat treatment time for all samples.

In general, the hardness of both alloys tends to decrease as the treatment time increases. However, series 1 has a slightly greater hardness, except when a 100-h aging treatment is applied.

According to Hamoud [18], Mg is an element that when alloying with Al can reduce some mechanical properties, in Al alloys, due to its chemical and structural properties. This behavior is shown in Figure 6, were the hardness tends to diminish as a function of the heat treatment time. It also can be observed that the alloy of series 2 shows the lowest hardness and is related with the highest Mg content. Furthermore, as the solubilization time increases, the alloy tends to build-up its grain size, causing a reduction of its mechanical properties [19]. Due to the experimental compositions, the elements used in both alloys (series 1 and 2) are susceptible to undergo contrary effects on the reduction of their mechanical properties as a consequence of the aging treatment time [20]. The main factors that particularly affect the phase dissolution are the size and distribution of the precipitates. It is worth mentioning an according to structure parameters such as: electronegativity, atomic radius, valence electrons, and atomic structure affinity, the alloying elements used shows a higher solubility capacity following the next in order: Al-Mg, Al-Si, and Al-Fe [21]. Then, $Mg_2Si$ phase shows the highest dissolution capacity, because it has more Mg, followed by the τ-AlMgZn and finally the $Al_3Fe$ phase with the lowest dissolution affinity. A new precipitation is observed after 5 h of aging treatment, causing a reduction of the alloy hardness value [22]. This precipitation leads to an incoherence with the matrix, which limits the dislocations movement along the phase. If the intermetallics tend to grow, the phases distributed in homogeneous form will be reduced inducing the alloy average hardness to decrease [23].

### 3.3. Electrochemical Tests

Electrochemical tests consisted on producing the polarization curves for all samples, both in the anode and the cathode branches. The polarization curves obtained from the analyzed samples are presented in Figure 6 and Table 6.

As observed in Figure 7 and Table 6, the alloys of series 2 exhibit more cathodic corrosion potentials ($E_{corr}$), with values of −1291.30 mV and −1215.69 mV, under the conditions of 0 h and 5 h respectively. These are the only conditions with a certain tendency to be passivated. In the as-cast condition (0 h) from series 2 showed a pseudo-passivation area in the anodic branch of the polarization curves; suggesting that a non-protective film is formed on the surface of the alloy. The anodic dissolution of the material decreases considerably, before reaching the point of approximately −997.38 mV, in which continuous dissolution carries on. Moreover, in the 5 h condition, polarization curves exhibit a passive zone that starts at the passivation potential, $E_{Pass}$ = −1162.81 mV, and breaks at the pitting potential,

$E_{pitt} = -1012.73$ mV. These two conditions are not suitable to be used in the manufacture of sacrificial anodes, since one of the properties a sacrificial anode must have is that it should not form passivating or protective films and should present a uniform corrosion. This mechanism is characteristic of Mg-Al alloys [24]. Thus, the following discussion will only focus on the behavior of the conditions where passivation does not occur.

**Table 6.** The electrochemical values obtained from the polarization curves of the AlMgZn alloys.

| Condition | $E_{corr}$ (mV) | Log $|i|$ (mA/cm$^2$) | $E_{pass}$ (mV) | $E_{pitt}$ (mV) |
|---|---|---|---|---|
| Base | −1046.01 | −0.11 | | |
| **Series 1** | | | | |
| As-Cast | −972.30 | −0.25 | | |
| 5 h | −935.39 | 0.32 | | |
| 24 h | −934.24 | 0.32 | | |
| 100 h | −1053.66 | −0.20 | | |
| **Series 2** | | | | |
| As-Cast | −1291.30 | −2.52 | −1189.53 | −997.38 |
| 5 h | −1215.69 | −2.86 | −1162.81 | −1012.73 |
| 24 h | −1012.73 | 0.19 | | |
| 100 h | −1023.80 | −0.05 | | |

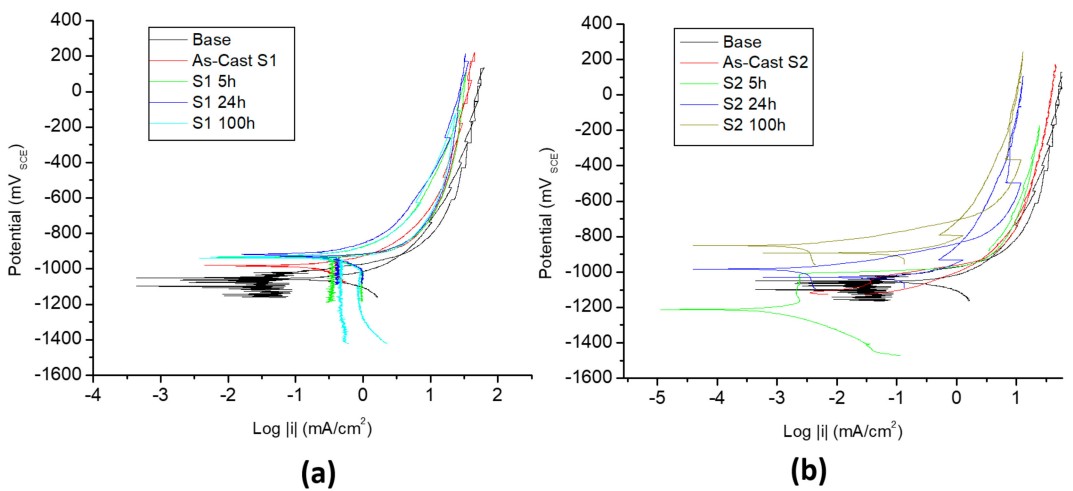

**Figure 7.** Polarization curves of the AlMgZn alloys (**a**) series 1 (**b**) series 2.

For series 1, conditions of 5 h and 24 h show the highest levels of corrosion current density (log i), with values 0.32 mA in both cases. Additionally, these conditions also exhibit the most anodic values of $E_{corr}$, i.e., −935.39 mV and −934.24 mV, respectively. These two conditions have a very similar behavior.

Furthermore, the dissolution potential ($E_{corr}$) must be negative enough to polarize the steel structure, which being the metal usually protected is the object intended for the application of the different alloys under study. Nonetheless, the potential should not be too negative, as this results in an unnecessary waste of power. The criterion of minimum practical protection and dissolution potential is based on the potential-pH Pourbaix diagram for iron, in which a well-defined immunity area is observed at −1041 mV$_{SCE}$, around pH 7, which corresponds to the saline solution used in this study. Likewise, in the same Pourbaix diagram, the maximum potential recommended is between −1200 mV$_{SCE}$ and −1280 mV$_{SCE}$, since by allowing more negative values, the risk of overprotection increases, generating an excessive reduction of hydrogen gas by the cathodic reaction. Alloys that meet these parameters in series 2 are the conditions of 24 h and 100 h, and for series 1 the condition of 100 h, in which the $E_{corr}$ observed was −1012.73 mV, −1023.8 mV, and −1053 mV, respectively. Furthermore, these three alloys have the lowest log i values (Figure 8), and show no tendency to be passivated.

Accordingly, the alloy most likely to be used as a sacrificial anode is the one of 100 h from series 1, and the ones of 100 h and 24 h from series 2.

Finally, as shown in Figure 7, in the polarization curves the alloy Base on the cathodic branch has a very unstable behavior. A behavior that does not allow it to reach the $E_{corr}$ before several events of anodic dissolution and intermittent equilibrium in the corrosion reactions occur. This behavior is most likely related to the phases present and the galvanic couples formed between them. Despite the above performance, $E_{corr}$ reaches equilibrium at -1046 mV, and this condition presents an $i_{corr}$ of $-0.11$ mA/cm$^2$. With these characteristics, this alloy meets the required attributes to be used as a sacrificial anode; however, due its cathodic behavior it is not recommended.

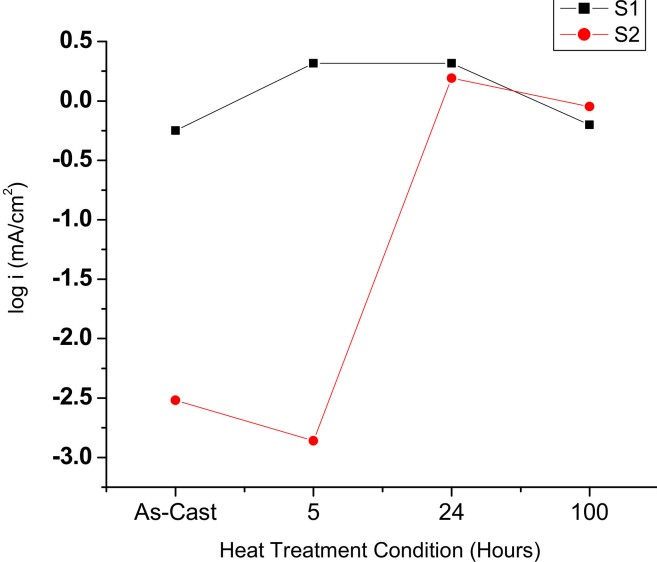

**Figure 8.** Current density as a function of heat treatment time.

### 3.4. SEM Observations after the Corrosion Tests

Once the electrochemical tests were concluded, samples were observed by scanning electron microscopy to determine the different morphologies obtained after the corrosion tests were carried out; microanalysis to corrosion products formed was performed as well.

Figure 9 corresponds to series 1. In Figure 9a the general appearance of the sample after the test is observed. Even though surface residues are clearly appreciated at these magnifications, if a selective attack is evident in regions of the intermetallic phase ($\tau$-AlMgZn and Mg$_2$Si), in the image this is seen as a gray phase bordering the dendritic structure. In Figure 9b the detail of this attack is illustrated, where a portion of a dendrite is surrounded by the interaction of the $\tau$ phase and the eutectic point, forming agglomerates and gaps. The average diffraction pattern obtained clearly indicates the formation of Zn, Mg, and Al oxides and chlorides; also, a sodium peak is appreciated, which is part of the solution used for the corrosion test (Figure 9c).

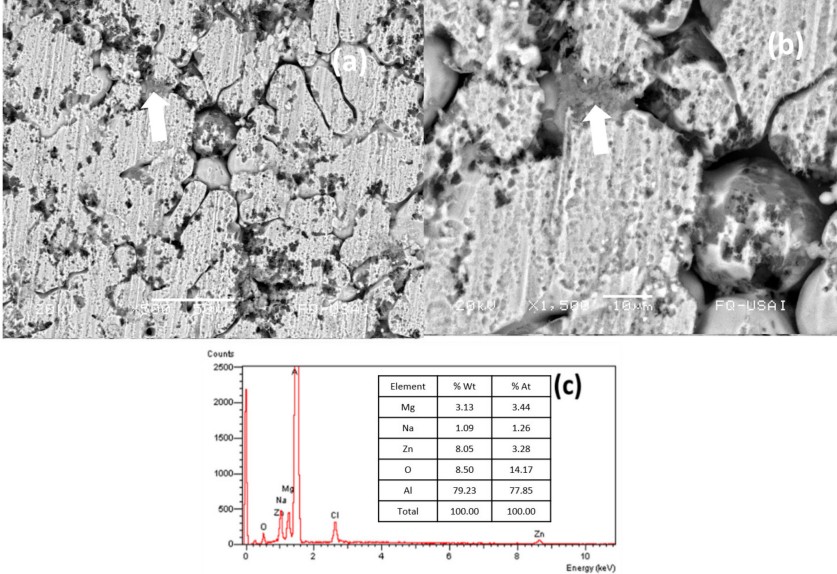

**Figure 9.** Micrographs and EDS recorded when stopped at a fixed current density (**a**) low magnification, (**b**) high magnification, and (**c**) microanalysis of the corrosion products from series 1.

Figure 10 corresponds to the samples of series 2. In Figure 10a a general aspect with no residues on the surface is observed; however, a selective attack is shown, even more severe than the one for the sample of series 1 (see Figure 9a). The latter is shown in Figure 10b, which details the attack on the intermetallic phase and shows that corrosion residues remain between the $\alpha$-phase (matrix), forming part of the eutectic point. Lastly, Figure 10c represents the pattern of the average diffraction obtained from the corrosion residues, which are characteristic of Zn, Al, and Mg chlorides and oxides; sodium used to prepare the solution for the corrosion tests is also present.

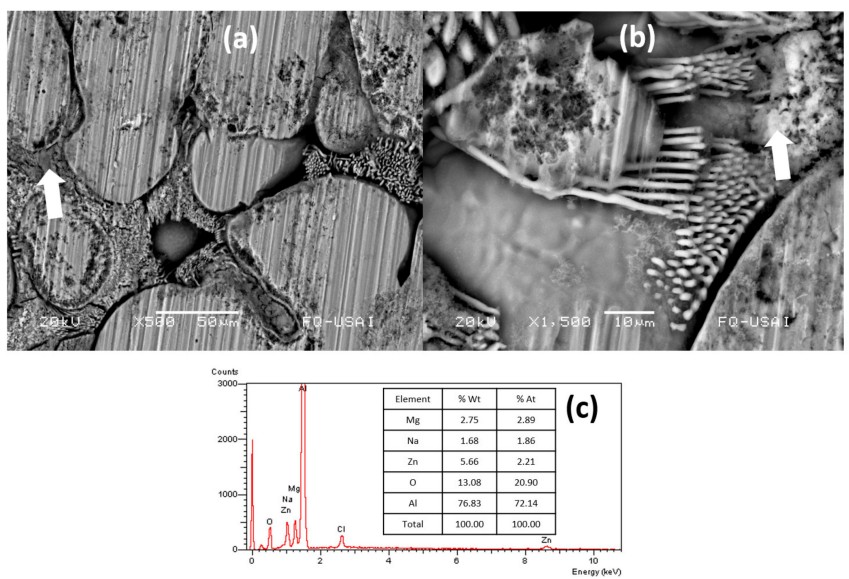

**Figure 10.** Micrographs and EDS recorded when stopped at a fixed current density (**a**) low magnification, (**b**) high magnification, and (**c**) microanalysis of the corrosion products from series 2.

Both series have galvanic couples formed between the $Mg_2Si$ and the $\tau$-AlMgZn phases with the $\alpha$-matrix, which is consistent with the electrochemical potential [25,26], as Mg has the lowest potential or is the most electronegative (−2.37 V), compared to other elements. For example, the $Mg_2Si$ phase is rich in Mg (Figure 3); therefore, this is the first phase to start dissolving. After the dissolution process

of $Mg_2Si/\alpha$, the $\tau$-AlMgZn/$\alpha$ phase starts dissolving until it disappears, as seen in Figures 9 and 10. When the eutectic of $\tau$-AlMgZn/$\alpha$ [27] corrodes, only parts with a tube morphology of the $\alpha$-phase (matrix) are left. This is more obvious in series 2, which has larger sizes and percentages of the phases present (see Tables 3 and 4). In general, the morphology of the samples of series 2 has a more localized corrosion, since the phases present are larger. By the heat treatment performed, the percentages of the phases present tend to diffuse the elements with more negative electrochemical potential in the matrix, which causes an increase in the reaction area between the galvanic couples. This is confirmed in the samples with the highest heat treatment times; which showed a phase percentage decrease between 20% and 30%, approximately. These same samples had the most negative $E_{corr}$ and the lowest log i, as a result of the diffusion of the elements with more negative potentials in the matrix, such as Zn and Mg; enhancing the corrosion of the surface and preventing the alloy passivation [17]. Another important issue to consider is the distribution of the same phases. In the case of series 2 the phases are larger, but closer to each other, so the diffusive range while performing the heat treatment is higher, since it has a higher concentration of elements with higher negative reduction potential. The heat treatment times affect the $Mg_2Si$ phase, this was more obvious in the alloys of 24 h and 100 h from series 2, and also, in the sample of 100 h from series 1, all of which showed the largest sizes of the corresponding phase.

## 4. Conclusions

Using specific conditions of thermal treatments, the mechanical and electrochemical properties of two alloys were modified: Series 1 (AlMg7.9-Zn13.3) and series 2 (AlMg10.5-Zn8.6). It was possible to control the quantity and size of the phases located in the grain boundaries and within the matrix. The main phases recognized were: $\tau$-AlMgZn, $Al_3Fe$, and $Mg_2Si$. The $\tau$-AlMgZn phase was the phase with greater proportion of precipitation in both alloys. As the treatment time increased, the phases decreased from 18% to 64%.

According to the electrochemical tests, it was found that the samples of series 1 (AlMg7.9-Zn13.3), are alloys with potential ranges of $-934.24$ to $-1053.66$ mV and current densities between -0.6310 and 2.0701 mA/cm$^2$, while the samples of series 2 (AlMg10.5-Zn8.6) showed potentials between $-1012.73$ to $-1291.3$ mV and current density of -0.0014 to 1.5488 mA/cm$^2$. These values are within the range expected to be used in the cathodic protection of steels.

Furthermore, samples of series 2 with 24 and 100 h of heat treatment are the best combination obtained. Having corrosion resistance, current density and micro-hardness can be used as potential sacrificial anodes, without the need to use alloying elements of higher cost. On the other hand, samples of series 1, are alloys with higher corrosion ranges, combined with a localized corrosion.

As discussed above, samples with an $E_{corr}$ within the most negative ones, the lowest log i, and with hardness above the one of the base alloy such as the samples of series 2 with 24 h and 100 h may be used as sacrificial anodes.

**Author Contributions:** Conceptualization, B.C. and A.H.; methodology, A.H.; software, A.H., Á.T. and S.S.; validation, B.C., S.S and J.M.; formal analysis, A.H., Á.T. and S.S..; investigation, A.H., Á.T., J.M. and S.S.; resources, B.C. and S.S.; data curation, A.H., Á.T.; writing—original draft preparation, A.H.; writing—review and editing, A.H. and B.C.; visualization, B.C.; supervision, B.C.; project administration, B.C.; funding acquisition, B.C.

**Funding:** This research received no external funding.

**Acknowledgments:** We would like to thank B. Eng. Iván R. Puente Lee for obtaining the SEM micrographs.

**Conflicts of Interest:** The authors declare no conflict of interest. The funders had no role in the design of the study; in the collection, analyses, or interpretation of data; in the writing of the manuscript, or in the decision to publish the results.

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
