# Peer review of "Effect of Experimental Aging on the Performance of Al-7.9Mg-13.3Zn and Al-10.5Mg-8.6Zn Alloys as Sacrificial Anodes"

_metals, doi:10.3390/met9080863_

Round 1

Reviewer 1 Report

Metals - Manuscript ID: metals-533260

This is an interesting and well written paper that is focused on the experimental aging of two Al-Zn-Mg sacrificial anodes. I recommend that corrections be made or considered before the paper is accepted for publication.

On  Line 29, a reference should be provided for the sentence: According to      Subodh, 45 million tonnes of aluminium…

Line  93, it would help the reader if the chemical compositions of the Keller’s and Graff-Sargent reagents were provided.

On lines 97 -102, the scan rate used in the polarisation curves should be      provided.

Lines, 161-172 and Tables 4 and 5, the discussion is centred on the % decrease in the phases. However, at 100 h the percentage of each phase increases  compared to the 5 h and 24 h and in some cases the as-cast sample. This is not discussed and the authors should provide some explanation to account for these observations.

Figure 7, why did the authors polarise the samples to 200 mV vs. SCE, given that  the Ecorr value is in the vicinity of -1000 mV and the pitting potentials are at similar values.

Table 7, use a consistent number of significant figures, e.g., Ecorr values of -1046 mV compared to -1012.73 mV.

The  micrographs and EDX plots in Figs. 9 and 10 are dominated by corrosion products, due to the fact that the electrodes were polarised to 200 mV vs SCE. This potential will never be reached when these anodes are used as sacrificial anodes. It would be more appropriate if the micrographs were      recorded after a period under open-circuit conditions, or after the      polarisation plots were recorded but stopped at a fixed current density, and      at potentials not too far from the corrosion potentials.

Lines 312 and 314 ….. current densities between -0.2 to 0.316 … of -2.86 to 0.19 mA. These are not current densities.  Convert to current density, units are mA/cm2 and use the same number of significant figures.

Author Response

Hello, 

Good evening!

I hope I can answer to you:

About Lines 161-171, of the decrease in the phases: The explanation of the changes in the quantity of the phases present in lines 161 to 172, is given in line 206 and later on, in order to understand it along with the hardness changes caused by the heat treatments applied.

About Figure 7: A higher voltage potential was applied in the anodic zone, only to corroborate that there were no unexpected changes in its behavior, not because it is necessary to apply such a high voltage.

The rest of the observations helped me a lot. I made the recommended changes.

best regards

Reviewer 2 Report

The manuscript is very well written with some interesting aspects not the aluminium corrosion protection. Some improvement should be made to the paper and then I believe could be accepted to this journal. The comments for improvement are in the file. 

Author Response

Hello, Good evening!

I hope I can answer to you:

About the area in line 105: Because of a measurement error in one of the cells, which are manufactured at our university so they have a percent error. However, the cells are usually manufactured with an area of 1 squared cm.

The rest of your comments, modify them in the rest of the writing, I hope it is to your liking.

I thank you in advance for your attention.

Best regards

Round 2

Reviewer 1 Report

The authors have addressed all the issues raised in the review process and the paper is now suitable for publication.

Author Response

Dear Reviewer:

I hope you are well. Allow me to thank you for your valuable comments and observations. Changes made according to what you indicated in the attachment. Please see the attachment.

Best regards
